# Limits to Reservoir Learning

## Abstract

In this work, we bound a machine's ability to learn based on computational limitations implied by physicality. We start by considering the information processing capacity (IPC), a normalized measure of the expected squared error of a collection of signals to a complete basis of functions. We use the IPC to measure the degradation under noise of the performance of reservoir computers, a particular kind of recurrent network, when constrained by physical considerations. First, we show that the IPC is at most a polynomial in the system size $n$, even when considering the collection of $2^n$ possible pointwise products of the $n$ output signals. Next, we argue that this degradation implies that the family of functions represented by the reservoir requires an exponential number of samples to learn in the presence of the reservoir's noise. Finally, we conclude with a discussion of the performance of the same collection of $2^n$ functions without noise when being used for binary classification.

## 1 Introduction

Reservoir computers(Frate et al. (2021); Mujal et al. (2023); Nakajima & Fischer (2021); Ganguly et al. (2017); Gonon & Ortega (2020); Duport et al. (2016); Martínez-Peña & Ortega (2023); Martínez-Peña & Ortega (2022); Verstraeten et al. (2007); Martínez-Peña & Ortega (2022)) are a particular kind of recurrent neural network where the only trained parameters are outgoing weights forming a linear layer between the internal parameters of the network, called the reservoir, and the readouts. This architecture drastically simplifies the process of training the network while maintaining high computational power. A defining aspect of a reservoir computer is its ability to perform inherently temporal tasks, such as time-series prediction or pattern recognition within sequences(Dominey (1995)). In this framework, the reservoir serves as a temporal kernel(Dong et al. (2020)), transforming the input sequence into a high-dimensional state represented in the hidden state of the reservoir. These hidden states are then linearly combined by trainable output nodes. The high-dimensional, temporal representation of data in the reservoir gives the output nodes enough flexibility to extract complex, non-linear features while avoiding the common issue of vanishing/exploding gradients or overfitting. Notably, due to the simple linear nature of the learned part of the network, the behavior and outcomes of reservoir computers are defined primarily by the reservoir. This interpretability sets reservoir computers apart from other deep learning approaches, making it valuable for applications requiring transparency or insight into the learning process. A crucial advantage is that the nonlinearities needed for learning are encapsulated within the physical dynamics of the reservoir, suggesting that we can leverage physical understanding to glean insights into the learning.

In this work, we prove several results about stochastic reservoir computers that are subject to physically motivated constraints. We find that stochasticity substantially impairs the learning capabilities of a physical reservoir computer. Specifically, what we find is that when considering the $2^n$ real-valued functions formed by the pointwise products of the $n$ real-valued functions forming the output signals of the reservoir, a particular measure of computational power called the information processing capacity (IPC)(Dambre et al. (2012); Hu et al. (2022); Polloreno et al. (2023); Polloreno (2023)) for any stochastic reservoir is at most polynomial in $n$ when the reservoir dynamics are constrained to certain physicality requirements. This is to be contrasted with the deterministic case, where the space of $2^n$ output functions generally trivially gives an exponential amount of IPC, even constrained by these requirements.

## 2 OVERVIEW

Often, when we consider a physical system, for instance thermodynamically or mechanically, we imagine that there is a prescribed algorithmic method for enlarging or extending the system. That is, when we say a reservoir, what we actually have in mind is a family of reservoirs, e.g. an interacting gas or a network of coupled oscillators. In this work, we will be interested in understanding how the performance of a computer built from one of these reservoirs changes with system size. In particular, because the number of degrees of freedom is increasing, we expect the system to be capable of increasingly better, or at least more interesting, computation. We will, however, establish a series of technical results that restrict the utility of reservoir computers constrained by physical assumptions.

In particular, we will consider stochastic reservoirs, which arise from, e.g., dissipation, incomplete information or measurement shot noise. In addition, we will assume the length of time the reservoir is run and the amount of energy the reservoir consumes are both polynomial in the system size $n$. This constrains the kinds of allowed operations - for instance, it is not possible to perform $n$-body operations (arbitrary long range interactions) in a single time step, since these require an exponential number of $k$-body ($k < n$) operations. Our first result, Theorem 2, will demonstrate that the dimension of the space we can perform regression onto can be meaningfully identified as polynomial, despite the exponential number of functions being considered. In particular, we will show that a large fraction of the signals produced by a reservoir are strongly dominated by noise, and that it requires a lot of waiting to distinguish one such signal from another. Our second result, Theorem 4 will give a characterization of the class of functions represented by the reservoir. Namely, due to a theorem of Bartlett et al.(Bartlett et al. (1994)), we will see that is possible to relate a certain kind of learnability of this class to its fat-shattering dimension (Bartlett et al. (1994)).

## 3 RESERVOIR COMPUTERS

A reservoir computer is a dynamical system, generally described by a system of differential equations, driven by an input $U(t) \in R^m$ and described by time varying degrees of freedom $X(t) \in R^n$ which represent its state. In this paper, we will consider systems consisting of $n$ degrees of freedom, which we will assume are bits. We note that this assumption is not particularly restrictive, and is in fact commonplace - any kind of physical dynamics can be encoded with vanishing error into a discretized signal. Because these degrees of freedom can be multiplied together to form new outputs, $Y(t) \in R^d$, in general the output dimension is $d \geq n$ and in this work we will consider $d = 2^n$ corresponding to all possible products of the $n$ signals. In the case of a stochastic reservoir computer, the state of the computer is in general a vector in the $2^n$ dimensional space of probability distributions over bitstrings of length $n$, i.e. $Y(t) = (p_{0...0}(t), ..., p_{1...1}(t))$. While it may be compelling to assume that the $n$ single-bit marginal distributions will contain the most computational utility, we note that arbitrary reservoirs can be used to construct complex and potentially highly correlated probability distributions. Thus, any argument that suggests there is a preferred collection of bitstrings among the $2^n$ possible ones is making further assumptions about the structure of the reservoir.

Typically, in a reservoir computing setting, one considers a dynamical system observed at discrete time-steps $t = 0, 1, 2, ...$, and the outputs are used to approximate a target function. Due to the general presence of memory in dynamical systems, we additionally define the concatenated $h$-step sequence of recent inputs $U^{-h}(t) = [U(t-h+1), U(t-h+2), ..., U(t)]$. While we may use the reservoir to learn a function of time, the reservoir's degrees of freedom themselves can be approximated by maps $x_k^h : U^{-h}(t) \mapsto R$. In particular, this is because we require that the reservoir satisfies the fading memory property. A dynamical system has fading memory if, for all $\epsilon > 0$, there exists a positive integer $h_0 \in N$, such that for all $h > h_0$, for all initial conditions, and for all sufficiently long initialization times $T' > h$, the $x_k(t)$ at any time $t \geq 0$ are well-approximated by functions $x_k^h$:

$$E((x_k(t) - x_k^h[U^{-h}(t)])^2) < \epsilon \tag{1}$$

where the expectation is taken over the $t + T'$ previous inputs(Polloreno (2023); Polloreno et al. (2023)). Due to two different sources of randomness in this paper, we next give definitions and notation for the different averages we compute, before we define the capacity of reservoir to reconstruct a signal.

**Definition 1.** *The input signal to a reservoir computer corresponds to a stochastic random variable and thus has an associated probability measure $\mu$, and we write $\overline{f}$ to denote averages of a quantity $f$ with respect to this measure, i.e.*

$$\overline{f} = \int d\mu f. \tag{2}$$

*A stochastic reservoir computer further has probabilistic dynamics, coming from, e.g., noise. Thus the outputs have an additional associated probability measure $\nu$, and we write $\langle f \rangle$ to denote the average of a quantity $f$ over this measure, i.e.*

$$\langle f \rangle = \int d\nu f. \tag{3}$$

**Definition 2.** *The capacity of a reservoir to reconstruct a signal $y(t)$ is given as*

$$C_T[y] = 1 - \min_\omega \frac{\sum_{t=1}^{T} (\hat{y}_\omega(t) - y(t))^2}{\sum_{t=1}^{T} y^2(t)}, \tag{4}$$

*where $\hat{y}_\omega$ is the estimate of $y$ produced from the reservoir.*

Typically $\hat{y}_\omega$ is produced via a linear weighting of the output signals, i.e. $\hat{y}(t) = \omega^T Y(t)$ for a weight vector $\omega$. Then, we can define the IPC as

**Definition 3.** *For a complete and countably infinite set of basis functions $\{y_1, y_2, \ldots\}$, the IPC of a dynamical system can be defined as*

$$\text{IPC} = \lim_{D \to \infty} \lim_{T \to \infty} \sum_{\ell}^{D} C_T[y_\ell] \le n. \tag{5}$$

(Hu et al. (2022) and Polloreno et al. (2023)) derive a closed form expression for the IPC of a stochastic reservoir, which we state here without derivation as Theorem 1. We can perform a spectral decomposition $\overline{\langle X \rangle \langle X \rangle^T} = VDV^T$ with positive definite, diagonal matrix $D$ and an orthogonal matrix $V$ to define the generalized noise-to-signal matrix $\tilde{\mathbf{Q}}_\xi$ as

$$\mathbf{I} + \tilde{\mathbf{Q}}_\xi \coloneqq D^{-\frac{1}{2}} V^T \overline{\langle XX^T \rangle} V D^{-\frac{1}{2}}, \tag{6}$$

where $\mathbf{I}$ is the identity matrix, so that $\tilde{\mathbf{Q}}_\xi$ describes the deviation of the reservoir from an ideal noiseless reservoir that produces orthogonal outputs.

**Definition 4.** *(Hu et al. (2022)) The right eigenvectors of $\tilde{\mathbf{Q}}_\xi$ are called the eigentasks of the reservoir.*

In (Polloreno et al. (2023)) it is shown that $\tilde{\mathbf{Q}}_\xi$ gives some measure of the reservoir stochasticity in the basis of the correlations imposed by the input signal and gets its utility from the following theorem:

**Theorem 1.** *(Hu et al. (2022); Polloreno et al. (2023)) The IPC is of a stochastic reservoir is given as*

$$\text{IPC} = \text{Tr}\left((\mathbf{I} + \tilde{\mathbf{Q}}_\xi)^{-1}\right)$$
$$= \sum_{k=1}^{n} \frac{1}{1 + \tilde{\sigma}_k^2} \le n, \tag{7}$$

where $\tilde{\sigma}_k^2$ are the eigenvalues of $\tilde{\mathbf{Q}}_\xi$. These eigenvalues correspond to noise-to-signal ratios, the inverse of signal-to-noise (SNR) ratios, of the reservoir at performing their respective eigentasks (see (Polloreno et al. (2023)) for more details). The reader may notice that this takes a similar form to the least squares solution to the linear regression problem with uncertainties on both the independent and dependent variables, and can be shown to come from similar considerations(Hu et al. (2022)). The outputs of the reservoir are, in general, post-processed depending on the learning task at hand. Because we conventionally optimize over linear weights, we are free to define the outputs of the reservoir up to a linear transformation without impacting the IPC. In particular, we will find particularly convenient the probability representation of the reservoir outputs.

**Definition 5.** *The probability representation of the outputs of a reservoir is given by the bitstring probabilities $p_k$, i.e. the output signal is given by $X(t) = (p_{0\ldots0}(t), \ldots, p_{1\ldots1}(t))$.*

Our results in this paper will be similar in spirit to the results in (Poulin et al. (2011); Shannon (1949)), which show that the space of states accessible by a physical computer, defined next, are exponentially vanishing in the total state space. Morally, these results suggest our ultimate result - how can physical states give rise to signals that have useful support on all $2^n$ basis vectors if the states themselves are exponentially vanishing?

**Definition 6.** *We define a stochastic reservoir as physical, motivated by (Poulin et al. (2011)), if its dynamics can be implemented in time polynomial in the system size, n. Furthermore, defining a $k-$body circuit element as a circuit element with $k$ inputs, we require that the dynamics of a physical stochastic reservoir be describable by circuits with $k-$body circuit elements, for some fixed $k$ independent of n.*

Intuitively, this definition rules out states that are not practically accessible to the reservoir. The $k$-body requirement stems from physicality constraints on the density of circuit elements. There are only so many circuit elements that can fit into a space and increasing this density arbitrarily with $n$, i.e. $n-$bit operations, is physically impossible. For the purposes of this paper we borrow the definition from (Poulin et al. (2011)) and provide a more detailed justification in Appendix A. In particular, we additionally prove a lemma that we will make later use of.

**Lemma 1.** *Consider the magnitude of the input to an $n-$bit reservoir computer, given as $u(t) = \|U(t)\|_2$. Then, the changes in probabilities $\frac{dp(u)}{du}$ are no more than polynomial in n.*

*Proof.* In Appendix A we argue from typical physical arguments that the changes in the probabilities $p(u)$ can only be changed polynomially rapidly in $u$. Furthermore, we argue in Appendix A that by construction of our definition of *physical*, we have only considered systems where $u$ is a polynomial in the system size $n$. □

## 4 STOCHASTIC RESERVOIRS HAVE SUBEXPONENTIAL CAPACITY

Generally speaking, the computational utility of a reservoir computer is fully characterized by the dimension of its externally observable dynamics. In the case of a deterministic reservoir, the state space is fully specified by $n$ bits. It is possible to further construct all $2^n$ functions on these bits, which then give the potential for $2^n$ capacity arising from correlations between the bits. (These new signals can of course fail to give additional IPC, for example consider a reservoir with outputs $f_1, f_2$ and $f_1 f_2$, with $\int d\mu(u) f_1 f_2 = 0$, where we have considered the standard $L^2$ inner product.) For example, the collection of polynomials $S$, given by $S = \{x, x^2, x^4, x^8 \ldots x^{2^n}\}$, where all $2^n$ elements of the powerset $2^S = \{\{\}, \{x\}, \{x^2\}, \ldots, \{x, x^2\}\ldots\}\}$ can be used to construct a collection $S'$ of exponentially many linearly independent polynomials by through multiplication, i.e. $S' = \{x, x^2, x^3 \ldots, x^{2^n}\}$.

As previously discussed, in the case of a stochastic reservoir, the state space is immediately naturally defined as $d = 2^n$ dimensional. However, despite the system requiring $2^n$ real numbers to be described, a natural question is if it is possible to utilize this $2^n$ dimensional space for useful computation, and in our case, learning? In particular we are able to construct $2^n$ signals by taking multiplicative products of the $n$ output signals - does this provide an exponential amount of IPC as in the deterministic case? We will find the answer is no, for any physical stochastic reservoir. As we will see, by introducing stochasticity, the performance of physical stochastic reservoirs is degraded to at most a polynomial in the number of output bits amount of IPC, even when considering all $2^n$ readout monomials (which form the conventional "state space" of the system). This makes it particularly important to be able to meaningfully select the "best" outputs, which requires some understanding of where the information is encoded. We will leave this problem to future work. In this section, we write only $p_k(u)$ to refer to the probability of bitstring $k$ at timestep $t$ when being driven by input $U(t)$, however the reader should be aware that because the reservoir has memory, it would be more appropriate to write $p_k(u) = p_k(U^{-a}(t))$ for some $a$ corresponding to the reservoir's effective memory, e.g. in Equation (1) $a = h_0$. We start with a lemma.

**Lemma 2.** (Hu et al. (2022)) *The IPC in the probability representation is given as*

$$\text{IPC} = \sum \frac{1}{1 + \tilde{\sigma}_k^2} = Tr\left(\Delta(\overline{\langle X \rangle})^{-1} \overline{\langle X \rangle \langle X^T \rangle}\right),\tag{8}$$

where $\Delta : R^d \to R^{(d,d)}$ *maps a vector to the diagonal matrix with entries given by the vector.*

*Proof.*

$$\mathrm{Tr}\left((\mathbf{I} + \tilde{\mathbf{Q}}_\xi)^{-1}\right) = Tr\left((\mathbf{D}^{-\frac{1}{2}}\mathbf{V}^T(\Delta(\overline{\langle\mathbf{X}\rangle}) - \overline{\langle\mathbf{X}\rangle\langle\mathbf{X}^T\rangle})\mathbf{V}\mathbf{D}^{-\frac{1}{2}} + \mathbf{I})^{-1}\right) \tag{9}$$

$$= \mathrm{Tr}\left((\overline{\langle\mathbf{X}\rangle\langle\mathbf{X}^T\rangle}^{-1}\Delta(\overline{\langle\mathbf{X}\rangle}))^{-1}\right) \tag{10}$$

$$= Tr\left(\Delta(\overline{\langle\mathbf{X}\rangle})^{-1}\overline{\langle\mathbf{X}\rangle\langle\mathbf{X}^T\rangle}\right) \tag{11}$$

where we have expanded the variance with respect to the reservoir stochasticity, taking advantage of the fact the signals are Bernoulli random variables in the probability representation. $\square$

**Theorem 2.** *The IPC of any physical stochastic reservoir is polynomial.*

*Proof.* The right hand side of Lemma 2, with the notation in Definition 5, gives

$$\mathrm{IPC} = \sum_k^{2^n} \frac{\int d\mu(u)p_k^2(u)}{\int d\mu(u)p_k(u)}, \tag{12}$$

for some measure $\mu$. For any value of $u$ we have that

$$\sum_k^{2^n} p_k(u) = 1. \tag{13}$$

Because of this, the signals must generally be relatively small, and moreover when they are not small, they must decay rapidly. For each $p_k(u)$ we imagine that it has behavior in these decaying regions, which we will call "tails", proportional to some $1/g_k(u)$, i.e. $p_k(u) \sim 1/g_k(u)$, so that we have the condition

$$\sum_k^{2^n} 1/g_k(u) = 1. \tag{14}$$

For instance, a constant number can have constant tails, a polynomial number can have polynomial tails and any super polynomial number needs to have inverse super polynomial tails. Because the functions can only grow polynomially by Lemma 1, we have that they have peaks that are $O(poly(u)/g_k(u))$. We have so far written all functions as functions of $u$, but we note that, as previously discussed in Lemma 1, the scale of the drive in any family of parameterized reservoirs will be related to $n$. Because we are integrating out $u$ below, we replace the functional dependence on $u$ with one on $n$. The IPC is thus bounded as given as

$$\mathrm{IPC} = \sum_k^{2^n} \frac{\int d\mu(u)p_k^2(u)}{\int d\mu(u)p_k(u)} \le \sum_k^{2^n} poly(n)/g_k(n) = poly(n), \tag{15}$$

where we have bounded each term based on the inequality $\int d\mu p_k^2(u) \le \frac{poly(n)}{g_k(n)} \int d\mu p_k(u)$, and used Equation (14). $\square$

## 5 CONNECTIONS TO LEARNING THEORY

In this section we discuss connections between the results proved in the previous section and modern ideas in statistical learning theory.

### 5.1 A LOWER BOUND ON THE FAT-SHATTERING DIMENSION

In the context of machine learning and statistical learning theory, complexity measures are used to characterize the expressive power of hypothesis classes and bound generalization error. One such complexity measure is the *fat-shattering dimension* (Kearns & Schapire (1994)), a concept that extends the classical VC Dimension (Shalev-Shwartz & Ben-David (2014)) to real-valued function classes, making it particularly suited for studying learning behavior of probabilistic classifiers and

regression problems. We will start by introducing the fat-shattering dimension, and a theorem of Barlett et al. (Bartlett et al. (1994)). We will use this theorem to prove Theorem 4 which states that the reservoir dynamics, i.e.

$$F' = \{p_{0\dots0}(t), \dots, p_{1\dots1}(t)\}, \tag{16}$$

are not agnostically learnable (defined below in Definition 10), and consequently have super polynomial fat-shattering dimension.

**Definition 7.** *Let $X$ be a domain of instances (i.e. an unlabeled data set) and let $\mathcal{H}$ be a class of real-valued functions mapping from $X$ to $[0,1]$, i.e., $h : X \to [0,1]$. Given a real value (the "width") $\gamma > 0$, the $\gamma$-fat-shattering dimension of $\mathcal{H}$, denoted by $\text{fat}_\gamma(\mathcal{H})$, is defined as the largest natural number $d$ for which there exist $d$ instances in $X$ and a set of thresholds $\{t_1, \dots, t_d\} \subseteq [0,1]$ such that for each subset $S \subseteq \{1, \dots, d\}$, there is a function $h_S \in \mathcal{H}$ satisfying the following conditions:*

- *For every $i \in S$, $h_S(x_i) \geq t_i + \gamma$.*

- *For every $i \notin S$, $h_S(x_i) \leq t_i - \gamma$.*

The fat-shattering dimension captures the ability of a hypothesis class to have a substantial gap of at least $2\gamma$ between the values assigned by certain hypotheses to elements inside subset $S$ and elements outside subset $S$. Including the thresholds and the real-valued range of the functions makes the fat-shattering dimension valuable when studying learning behavior for probabilistic classifiers and regression problems. In this work, we consider a reservoir that has $2^n$ possible outputs. We imagine using these outputs to perform a classification task on the input signal by considering a linear combination of empirical estimates $\hat{p}_i(u)$ to perform binary classification on $u$. As a particularly illustrative example, consider "switching signals" (e.g Figure 1b and Figure 1a) which, upon receiving $u_i$ with $i \in S$, raises $p_{0\dots0}(u_i)$ above 0.5 by at least $\gamma$ and with $i \notin S$ lowers $p_{0\dots0}(u_i)$ below 0.5 by at least $\gamma$. Choosing $t_i = 0.5$, such a reservoir has a fat-shattering dimension of at least $|S|$. Such a reservoir may not be implementable, however, given the specific dynamics available, or the details of the input signal. In particular, the illustrative example of "switching signals" (Figure 1b and Figure 1a) is limited by the ability of physical system to drive large enough changes in the dynamics to produce these signals. Furthermore, we have considered the deterministic case here, where the functions $p_k(u)$ are treated as accessible real-valued functions. To accurately model the stochastic signals considered in this paper, we must consider the setting where these real-valued functions are instead corrupted by noise. In particular, we will assume they are the parameters of a Bernoulli distribution. To this end, (Bartlett et al. (1994); Kearns & Schapire (1994); Kearns et al. (1992); Kearns (1998); Haussler (1992)) consider the model of probabilistic computation. To start, we define probabilistic concepts and agnostic learning.

**Definition 8.** (Kearns & Schapire (1994)) *A probabilistic concept $f$ over a domain set $X$ is a mapping $f : X \to [0,1]$. For each $x \in X$, we interpret $f(x)$ as the probability that $x$ is a positive example of the probabilistic concept $f$. A learning algorithm in this framework is attempting to infer something about the underlying target probabilistic concept $f$ solely on the basis of labeled examples $(x, b)$, where $b \in \{0, 1\}$ is a bit generated randomly according to the conditional probability $f(x)$, i.e., $b = 1$ with probability $f(x)$. The value $f(x)$ may be viewed as a measure of the degree to which $x$ exemplifies some concept $f$.*

To connect the capacity (Definition 2) to existing work in learning theory, we now define the error integral of a classifier.

**Definition 9.**

$$\text{er}_P(h) = \int |h(x) - y| dP(x, y) = \overline{\langle \sqrt{(h(x) - y)^2} \rangle}, \tag{17}$$

*denotes the error integral of a classifier $h$ trained on samples from a probability distribution $P(x, y)$, with the averaging notation being defined in Definition 1.*

The following definition requires the learner to perform almost as well, with high probability, as the best hypothesis in some class $G$, referred to as a *touchstone class*, for any particular learning task. The word agnostic in this setting is used because there is no assumption of an underlying function generating the training examples. We will consider a randomized learning algorithm which takes a sample of length $m$ and chooses sequences $z \in Z^m$ at random from $P_Z^m$, and gives it to a deterministic mapping $A$ as a parameter. Deterministic algorithms are a subset of these mappings where the $A$ ignores the random string.

**Definition 10.** (Bartlett et al. (1994)) *Suppose G is a class of $[0, 1]$-valued functions defined on $X$, $P$ is a probability distribution on $X \times [0, 1]$, $0 < \varepsilon, \delta < 1$, and $m \in N$. A randomized learning algorithm $L$ is a pair $(A, P_Z)$, where $P_Z$ is a distribution on a set $Z$, and $A$ is a mapping from $\bigcup_m (X \times R)^m \times Z^m$ to $[0, 1]^X$. For an algorithm $A$ and a distribution to be learned $D_Z$ on a set $Z$, we write that $L = (A, D_Z)$. We say that $L$ $(\varepsilon, \delta)$-learns in the agnostic sense with respect to $G$ from $m$ examples if, for all distributions $P$ on $X \times [0, 1]$*

$$\left( P^m \times D_Z^m \right) \{ (x, y, z) \in X^m \times [0, 1]^m \times Z^m :$$
$$\mathrm{er}_P(A(x, y, z)) \geqslant \inf_{f \in G} \mathrm{er}_P(f) + \varepsilon \} < \delta,$$

*where $\mathrm{er}_P(\cdot)$ is the error integral introduced in Definition 9. The function class $G$ is agnostically learnable if there is a learning algorithm $L$ and a function $m_0 : (0, 1) \times (0, 1) \to N$ such that, for all $0 < \varepsilon, \delta < 1$, algorithm $L(\varepsilon, \delta)$-learns in the agnostic sense with respect to $G$ from $m_0(\varepsilon, \delta)$ examples. If, in addition, $m_0$ is bounded by a polynomial in $1/\varepsilon$ and $1/\delta$, we say that $G$ is small-sample agnostically learnable.*

In our case, we will see that physical stochastic reservoir computers provide an example of a particular probability distribution for which learning the functions describing their dynamics is not small-sample agnostically learnable due to the degradation in capacity on those functions. We start with a corollary of Theorem 2 and a theorem by Bartlett et al., before proving our second theorem.

**Corollary 1.** *For any physical stochastic reservoir there are $\Omega(g(n))$ learning tasks $f_i$ such that*

$$\mathrm{er}_P(f_i) \geq 1 - O(poly(n)/g(n)), \tag{18}$$

*where $g(n) = \omega(poly(n))$.*

*Proof.* Because, for small errors ($|(h(x) - y)| \leq 1$),

$$\mathrm{er}_P(h) \geq \sqrt{\langle h^2 \rangle_T (1 - C_T[h])}, \tag{19}$$

we see a small capacity also implies a large error. This will allow us to prove our corollary and connect with the statistical learning literature. Specifically, we see that

$$\mathrm{er}_P(h) \geq \sqrt{\langle h^2 \rangle_T} - \sqrt{\langle h^2 \rangle_T} C_T[h]/2 + O(C_T[h]^2), \tag{20}$$

for small capacity.

In Theorem 2 we argued that for any physical stochastic reservoir there are super-polynomially many functions in the touchstone class, $f_i \in F'$ (defined in Equation (16)), that have inverse superpolynomially poor capacity, i.e. $h$ such that

$$C_T[h] = O(poly(n)/g(n)), \tag{21}$$

where $g(n) \in \omega(poly(n))$. We consider the learning problem with the touchstone class $F$

$$F = \{f_i \mid f_i \text{ is a linear combination of functions in } F' \text{ and } C_T(f_i) = O(poly(n)/g(n))\}. \tag{22}$$

Note that in particular, this class includes the poor SNR eigentasks of the reservoir. The IPC, as we have seen, gives a measure of the SNR over each signal, and is normalized by definition. Replacing the capacity in our previous inequality with $poly(n)/g(n)$, we have

$$\mathrm{er}_P(f) \geq \sqrt{\langle f_i^2 \rangle_T}(1 - O(poly(n)/g(n))). \tag{23}$$

These functions form an orthonormal basis, and so we set the norm of the function to one, arriving at the desired inequality. $\square$

This corollary relates the classification error of Definition 9 of a reservoir performing an eigentask to its capacity, defined in Definition 2. Intuitively this is possible because the capacity is a measure of the SNR, and low SNR makes classification more difficult. We now give a theorem of Bartlett et al. that relates the fat-shattering dimension to small-sample agnostic learnability, and follow with our own theorem, showing that the fat-shattering dimension of the probabilistic concept class of reservoir functions has super-polynomial fat-shattering dimension. We start with a technical definition.

**Definition 11.** (Bartlett et al. (1994); Haussler (1992)) *Consider a $\sigma$−algebra $\mathscr{A}$ on Z. A class of functions G is **PH**-permissible if it can be indexed by a set T such that*

1. *T is a Borel subspace of a compact metric space $\overline{T}$ and*

2. *the function $f : Z \times T \to R$ that indexes G by T is measurable with respect to the $\sigma$−algebra $\mathscr{A} \times \mathscr{B}(T)$, where $\mathscr{B}(T)$ is the $\sigma$−algebra of Borel sets on T.*

*We say a class G of real-valued functions is permissible if the class $l_G : \{l_g \mid g \in G\}$, $l_g : (x, y) \to (y − g(x))^2$ is **PH**-permissible.*

**Theorem 3.** (Bartlett et al. (1994)) *Suppose G is a permissible class of $[0, 1]$ valued functions defined on X. Then G is agnostically learnable if and only if its fat-shattering function is finite, and G is small-sample agnostically learnable if and only if there is a polynomial p such that $\mathrm{fat}_\gamma(G) < p(1/\gamma)$ for all $\gamma > 0$.*

We will now demonstrate that due to the degradation in IPC, a learning algorithm cannot, in general, differentiate between the different learning tasks described by a reservoir's eigentasks without using an exponential number of observations. Hence we will demonstrate that there does not exist a learning algorithm that can agnostically learn the reservoir dynamics with $poly(1/\delta, 1/\epsilon)$ samples. Specifically, we show that the assumption that the class of functions encoded by the dynamics of the reservoir is learnable in the presence of noise is not compatible with that class containing many orthogonal functions. If they are learned, they are learned despite the noise, and must all therefore be similar to a single learned function. But they cannot be too similar to the learned function, because then they would be similar to each other, and they are orthogonal.

**Theorem 4.** *There does not exist a polynomial p such that $\mathrm{fat}_\gamma(F) < p(1/\gamma)$ for all $\gamma > 0$ for the concept class of functions F corresponding to reservoir dynamics of an infinite family of reservoirs.*

*Proof.* Since the errors are one-sided - the learner cannot perform better than the reservoir function at its own eigentask - the condition for agnostic learnability, is that for all $\epsilon > 0$ with probability $1 − \delta > 0$, it is possible to take enough samples so that

$$\sum_i^{|F|} \mathrm{er}_P(A(x, y, z)) \le \sum_i^{|F|} \mathrm{er}_P(f_i) + \epsilon. \tag{24}$$

Using Corollary 1, we can relate the error to the capacity, so that for the collection of functions $F$ with small capacity, this is equivalent to

$$\sum_i^{|F|} \mathrm{er}_P(A(x, y, z)) \le \sum_i^{|F|} (1 − c_i + \epsilon), \tag{25}$$

with high probability, where $c_i$ denote the terms that are $O(poly(n)/g(n))$ in Equation (23). For small-sample agnostically learnability, we have the number of samples $m_0 = poly(1/\delta, 1/\epsilon)$. Because the identically zero function is among the functions that the learning algorithm must perform well on, i.e. $0 \in F$, we can bound the the probability of success based on the probability that the learning algorithm falsely reports the identically zero function. For the functions we are considering, the capacity is low, and hence from our proof of Theorem 2 we can choose $n$ such that the probability of the learning algorithm sampling anything nonzero is small. In particular, for a probability $q$ that any particular sample is nonzero, the probability $p$ that the learner samples all zeros, consistent with the function being the zero function, is given as

$$p = (1 − q)^{m_0} \approx m_0 q \approx poly(1/\delta, 1/\epsilon)poly(n)/g(n), \tag{26}$$

for $g(n) \in \omega(poly(n))$. Making this approximation requires $m_0 q \ll 1$, so that we choose $poly(n, 1/\epsilon, 1/\delta)/g(n) \ll 1$. A sufficient condition, therefore, is choosing $n \approx \max(1/\delta, 1/\epsilon)$. In this case, the learner is unable to do better than guessing that the signal is small or zero, and incurring an error proportional to the size of the signal. Hence the condition for agnostic learnability

becomes

$$
\begin{aligned}
\sum_i^{|F|} \mathrm{er}_P(A(x,y,z)) &= \sum_i^{|F|} \left( \int |f_i(x) - A(x,y,z)| dP(x,y) \right) \\
&\geq \sum_i^{|F|/2} \int |f_i(x) - f_{i+|F|/2}(x)| dP(x,y) \\
&\geq \sum_i^{|F|/2} \int (f_i(x) - f_{i+|F|/2}(x))^2 dP(x,y) = |F|,
\end{aligned}
\tag{27}
$$

where we have used the triangle inequality for the first inequality, $|f_i(x) - f_{i+|F|/2}(x)| \leq 1$ and $|f_i(x) - f_{i+|F|/2}(x)| \geq (f_i(x) - f_{i+|F|/2}(x))^2$ for the second inequality and orthogonality of the reservoir functions for the final equality. In particular, integrating over the input measure first, e.g.

$$
\int d\mu(x) d\nu(y) f_1(x,y) f_2(x,y) = \int d\nu(y) \int d\mu(x) f_1(x,y) f_2(x,y) = 0,
\tag{28}
$$

due to the orthogonality of the functions under $\mu$. Putting this together with Equation (25), we have

$$
|F| \leq \sum_i^{|F|} \mathrm{er}_P(A(x,y,z)) \leq \sum_i^{|F|} (1 - c_i + \epsilon).
\tag{29}
$$

so that this is violated if

$$
\frac{1}{|F|} \sum_i^{|F|} c_i > \epsilon
\tag{30}
$$

i.e. that the worst signals still provide some small utility on average. This requirement can be made arbitrarily weak since the condition for agnostic learnability is that this is true for all $\epsilon$, and hence what we have demonstrated is that there is an $(\epsilon, \delta, m_0)$ for which we can choose $n$ such that the reservoir dynamics are not small-sample agnostically learnable. An alternative interpretation is that any such learner would provide a witness that the eigentasks are not orthogonal under the reservoir dynamics, since this would require the eigentasks to be too similar to each other. Hence, we conclude from Theorem 3 that there is no polynomial $p$ such that $\mathrm{fat}_\gamma(F) < p(1/\gamma)$ for all $\gamma > 0$ if Equation (30) holds. $\qquad\square$

At this point we have taken a kind of limit, assuming that our reservoir is a member of an infinite family of reservoirs. For each $(\epsilon, \delta)$, we have required only that the number of samples scales polynomially with the reservoir size, and hence not only does the function class corresponding to the infinite family of reservoirs have a superpolynomial fat-shattering dimension, but also the functions that constitute this family are generally efficient (in the number of samples) to compute.

## 6 DISCUSSION

By connecting ideas from learning with dynamical systems to concepts in statistical learning theory, we have found that the fat-shattering dimension of the functions represented by reservoir dynamics is superpolynomial in the inverse of the fat-shattering width $\gamma$. Intuitively, what we have shown is that, because reservoirs have a large number of low SNR eigentasks, and because no learning algorithm can be expected to do better than a reservoir at its eigentasks while being subjected to the same noise as the reservoir, the class of functions represented by the reservoir is itself challenging to learn. Surprisingly, this informs us about the growth of the fat-shattering dimension of the model at small scales ($\gamma \to 0$), whereas considerations from the dynamics in Theorem 4 immediately rule out a collection of "switching" signals which instead seems to suggest a restriction on the growth of the function class at large scales ($1/\gamma \to 0$). While the fat-shattering dimension, similarly to the VC dimension(Shalev-Shwartz & Ben-David (2014)), can be further used to establish bounds on generalization error through connections with Rademacher complexity(Shalev-Shwartz & Ben-David (2014)), we leave this to future work.

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

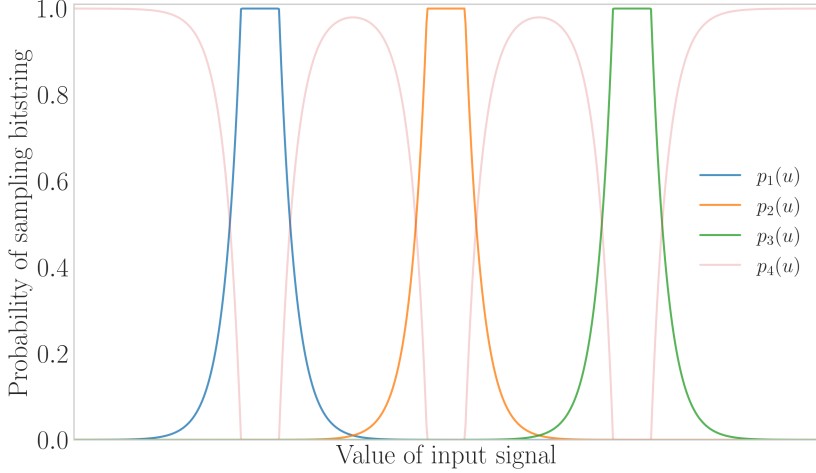

(a)

(b)

Figure 1: a) "Switching signals", with exponential tails, proportional to $2^u$. With exponential changes in the probabilities, the reservoir is able to drive very rapid changes in the output signals, and thus able to fit four signals with trivial confusion probabilities near their peaks. In particular, observe that the red, low alpha signal is able to achieve a probability of nearly one between the first and second (blue and orange) and second and third (orange and green) signals. b)"Switching signals", with polynomial tails, proportional to $u^2$. The overlap where the signals cross is more substantial, and consequently fewer signals can be fit into the space. This results in significant rates of "confusion", e.g. the red signal has probabilities that are significantly smaller than 1 in the regions between pairs of the first and second (blue and orange), or second and third (orange and green) higher alpha signals.

## A    Physical Reservoirs and Proof of Lemma 1

The idea of physical or physically accessible circuits have been discussed in the literature (e.g. (Shannon (1949); Poulin et al. (2011)), however we are not aware of references that prove that general stochastic digital computers can only have polynomial changes in the probabilities. We thus include a proof of Lemma 1 here. Our proof relies on an understanding of quantum computational dynamics, and thus has not been included in the main text, to avoid burdening the reader. In a future version of this manuscript we will instead use the formalism of generators of the dynamics of probabilistic circuits when considered as Markov chains. The current argument is included due to its familiarity to the author. In principle, an argument involving purely classical dynamics would be both simpler and more elucidating. In particular, in the same way that unitary operators have generating Hermitian operators, Markov chains have generators. Markov chains are equivalent descriptions of probabilistic circuits in the same way unitary operators are equivalent descriptions of quantum circuits.

*Proof.* (Poulin et al. (2011)) establishes that arbitrary time-dependent quantum dynamics are no stronger (and clearly as strong as, by taking logarithms of the gates and applying them as Hamiltonians) the quantum circuit model. We start by noting that quantum circuits can only generate polynomial changes in the probability amplitudes. To see this, all evolutions are given as

$$\dot{\rho} = i[H, \rho], \tag{31}$$

so that for real $H$ the solutions are given roughly as complex exponentials in the (real) eigenvalues of $H$. If we assume that the norm of $H$ is at most polynomial in the system size and time, this is sufficient to argue that the changes in the amplitudes are also at most polynomial in $n$ and $t$.

For quantum circuits we will consider circuits that consist of coherent AND (Toffoli) and OR gates, as well as $k$-bit rotations, i.e. Equation (32). Each of these circuits is bijective with a classical circuit consisting of classical AND, OR and $k$-bit probabilistic not gates via the following construction.

First, consider any function $p(t)$ describing the parameter of a time-varying Bernoulli distribution associated with a bit. This is identically given by averaging two over two unitary operations:

$$U_1(t) = e^{-i \arccos(\sqrt{p(t)})X} = \begin{pmatrix} \frac{\sqrt{p(t)}}{2} & -\frac{1}{2}i\sqrt{1 - p(t)} \\ -\frac{1}{2}i\sqrt{1 - p(t)} & \frac{\sqrt{p(t)}}{2} \end{pmatrix} \tag{32}$$

$$U_2(t) = e^{i \arccos(\sqrt{p(t)})X} = \begin{pmatrix} \frac{\sqrt{p(t)}}{2} & \frac{1}{2}i\sqrt{1 - p(t)} \\ \frac{1}{2}i\sqrt{1 - p(t)} & \frac{\sqrt{p(t)}}{2} \end{pmatrix} \tag{33}$$

$$\hat{p}(t) = \frac{1}{2}(U_1^*(t) \otimes U_1(t) + U_2^*(t) \otimes U_2(t)), \tag{34}$$

where $\hat{p}(t)$ describes the conjugation action on the vectorized density matrix, and follows from Roth's lemma ((Horn & Johnson (2012); Ward (1999), i.e.

$$\hat{p}(t) \operatorname{vec}(\rho) = \frac{1}{2}(U_1(t)\rho U_1^\dagger(t) + U_2(t)\rho U_2^\dagger(t)), \tag{35}$$

where, given a density matrix $\rho$ of size $n \times n$, the vectorization of $\rho$, denoted by vec($\rho$), is an operator that rearranges $\rho$ into a column vector of size $n^2 \times 1$. This is done by stacking the columns of $\rho$ on top of one another. Mathematically, the vectorization operation gives

$$
\rho = \begin{bmatrix} \rho_{11} & \rho_{12} & \cdots & \rho_{1n} \\ \rho_{21} & \rho_{22} & \cdots & \rho_{2n} \\ \vdots & \vdots & \ddots & \vdots \\ \rho_{n1} & \rho_{n2} & \cdots & \rho_{nn} \end{bmatrix}, \quad \mathrm{vec}(\rho) = \begin{bmatrix} \rho_{11} \\ \rho_{21} \\ \vdots \\ \rho_{n1} \\ \rho_{12} \\ \rho_{22} \\ \vdots \\ \rho_{n2} \\ \vdots \\ \rho_{1n} \\ \rho_{2n} \\ \vdots \\ \rho_{nn} \end{bmatrix} \tag{36}
$$

To generate correlated changes in probabilities of different bits, we can exponentiate, e.g., $X \otimes X$, which generates transitions between 00 and 11. This argument thus establishes an equivalence between dynamics in a classical stochastic circuit, and the dynamics of two different quantum circuits.

Now that we have established a bijection, we would like to argue that the rate of change of the two different dynamics are polynomially related. To see this, consider that any amplitude $\alpha(t) \sim \sqrt{p(t)}$, so that

$$
\frac{d\alpha(t)}{dt} \sim \frac{1}{2\sqrt{p(t)}} \frac{dp}{dt}. \tag{37}
$$

By assumption, the left hand side is a polynomial, so that

$$
\frac{dp}{2\sqrt{p(t)}} \sim poly(t)dt \tag{38}
$$

$$
\sqrt{p(t)} \sim poly(t), \tag{39}
$$

and we are done. In particular, we see that if a polynomial depth k-local classical stochastic circuit produces superpolynomial changes in the bitstring probabilities, there must exist a polynomial depth k-local quantum circuit which produces superpolynomial changes in amplitudes, which is impossible. $\square$

This argument has shown that arbitrary $p(t)$ can be produced by creating two quantum systems, and that a superpolynomial change in $p(t)$ would produce a superpolynomial change in a quantum circuit. The complexity of the classical circuit that produces $p(t)$ is related to the complexity of the quantum circuit that produces $U_1(t)$ and $U_2(t)$, and in particular can generate a change in probabilities $p(t)$ that is superpolynomial if and only if the corresponding quantum circuit generates a change in the amplitudes of the corresponding quantum state that is superpolynomial.

