# OpenReview forum: "Limits to Reservoir Learning"
_ICLR.cc/2024/Conference — Submitted to ICLR 2024_

### Official Review · Reviewer_wi5q · 2023-10-31

**Soundness:** 2 fair
**Presentation:** 2 fair
**Contribution:** 2 fair
**Rating:** 3
**Confidence:** 2

**Summary:**

This paper characterizes the Information Processing Capacity (IPC) of
physically-constrained reservoir computers and establishes that IPC is
subexponential. This finding is interpreted in terms of bounds on the
fat-shattering dimension of learning tasks achieved with reservoir computers.

**Strengths:**

This paper obviously took a great deal of domain expertise and original, careful
thought.

I am unable to sufficiently evaluate all of the technical details of this paper,
but I was able to follow the major steps with significant effort and the technical
details appear sound.

**Weaknesses:**

I spent quite a lot of time reviewing this paper, and despite the fact that the
prose is well-polished, the high-level ideas required reading referenced papers
and to make sense of,  and I won't claim that I finally achieved satisfactory
understanding. Please refer to the "Questions" section of my review for detailed
feedback and questions.

The significance of the findings is somewhat dubious relative to the claims that
the authors "bound a machine's ability to learn", made in the abstract (For
comparison, I would claim that the Bekenstein bound equally "bounds a machine's
ability to learn based on computational limits implied by physicality"; each
claim requires enough equivocation as to feel vacuous by the end.), but a more
nuanced accounting of the results are given in the discussion.

I am willing to revise my evaluation, but for the effort I put into
understanding this paper, I did not feel like I came away with new insights
or appreciations.

**Questions:**

I will detail some of my confusions (and my resolutions to these confusions below) before asking more pointed questions.

In Dambre et. al.'s work, the capacity $C_T[y]$ is defined with respect to the
function (rephrased using the present notation) $y(t) = z(U^{-h}(t))$, where $z$
is some real-valued function *on the input history*, and the vector of
observables $Y$ is *implicitly determined by $U^{-h}(t)$* (These bold phrases
were not clear upon an initial reading of this paper). Intuitively, the capacity
is a measure of the ability reconstruct the function $z(U^{-h}(t))$ using some
linear combination of observables $Y$ over $T$ steps.

Since $C_T[y]$ is *data-dependent*, this quantity makes sense in general only if
we some probability distribution over $U^{-h}(t)$ that we have defined (Dambre
et. al. consider i.i.d. $u(t)$, and the present paper defines input as a
stochastic random variable). In Equation 8 of the referenced work, this explicit
dependence on data implies that the definition of IPC is such that the capacity
is summed over different functions *on the same data* (this fact is not made
immediately clear in the present work).

I was also initially confused by the lack of clarity regarding the inner product
between "basis functions" $y_l$: this inner product technically depends on the
dynamics of the system and the inputs, as noted by Dambre et. al, while the
present work appears to elide this point. The value of $n$ in Equations (5) and
(7) further confuses this issue, since $n$ was earlier used to represent the
number of internal degrees of freedom, not the number of orthogonal observables
of the system.

A major question I stumbled on was why we should believe that it is possible,
even in the absence of noise, for IPC of $2^n$ using the products of $n$
independent, bit-valued degrees of freedom. I did not and still do not
understand the example given in the first paragraph of Section 4, but I believe
I partially resolved this issue for myself:

I understand that each $x_{k} \colon U^{-h}(t) \to {0, 1}$ for $k \in [n]$
represents a single degree of freedom of the system. I also understand that
$U(t)$ is formulated as continuous and $m$-dimensional. In the limit of $h \to
\infty$, it makes intuitive sense that the $2^n$ products of the $n$ degrees of
freedom could be reliably distinguished in the case that the expected dynamics
of the system provide for unique probability densities for each combination of
bits.

## Q1
My remaining question on this point is whether deterministic systems subject to
the constraints of Definition 6 (characterizing circuit complexity of the
system) also have exponential capacity. That is, is it inherently the
stochasticity that limits the IPC of the system, or is it the limits on
dynamical complexity, or are both required?

## Q2
If the central claim is that the introduction of *noise* reduces IPC to
subexponential while the deterministic case is exponential (with or without the
circuit complexity assumption) I would expect a clear limiting case to be
established by which the limit of zero noise in the stochastic setting recovers
exponential capacity. Is this limiting case evident from the results of this paper?

## Q3
Finally, as a separate question, are the authors aware of treatments of
reservoir computers that relate IPC to characterizations of the underlying
dynamical system via the Koopman operator? Intuitively, I get the impression
that there are connections here that may be useful.

---

### Official Review · Reviewer_CfFW · 2023-11-01

**Soundness:** 3 good
**Presentation:** 3 good
**Contribution:** 2 fair
**Rating:** 6
**Confidence:** 3

**Summary:**

In this paper the authors present an analysis of reservoir computing systems from the point of view of learnability. The authors first define reservoir a system of multiple predictors linearly combined to give the final result. The authors show that the capacity of a reservoir classifier have an quadratic capacity in the ability of discriminating from a dataset of $2^n$ size. In addition the authors describe the finding the framework of the fat-shattering dimension which is an interesting addition.

**Strengths:**

The paper framework looks solid. The approach of breaking the sequential nature of reservoir computing to classification is a good reduction mechanism for analysis.

The fat-shattering link and analysis is very helpful in understanding the overall reasoning of the authors and of the conclusion

**Weaknesses:**

Under the assumption that tin the classification task (also see questions) the result is a linear concatenation of the outputs of the elements in the reservoir is it surprising that the system has a quadratic capacity?

I am not sure if I completely understood the link between the sequence prediction and the classification within the limits of the framework. The original formulation of reservoir framework is for sequential signals, that might benefit from the temporal relations between the set of consequent chains of bit of length $2^n$. Thus I am not sure if the analysis hold for both problems.

**Questions:**

So if I understood correctly the studied model is a linear combination of individual outputs form the reservoir and this linear combination poses the limit on the number of distinguishable classes by the whole system.

Therefore the first question that I have is that what is the reservoir is reduced to a single element and the output is not a concatenation but a single prediction? Does the analysis holds or because this is an agnostic method it does not apply?

A second question related is that, if the original model is again reduced to a single prediction/classification it can be compared to a Markov chain model. Therefore is the limitation of the linear combination surprising or novel enough?

---

### Official Review · Reviewer_AaBM · 2023-11-01

**Soundness:** 1 poor
**Presentation:** 1 poor
**Contribution:** 1 poor
**Rating:** 1
**Confidence:** 4

**Summary:**

The work investigates properties and limitations to reservoir computing methods. In particular the work looks at the capacity to reconstruct signals, stochastic reservoirs, and connections to learning theory.

**Strengths:**

Reservoir computing is an efficient approach for learning dynamical systems, and the work aims to provide more insights into this class of methods.

**Weaknesses:**

IPC is not a standard measure for information processing capabilities of artificial neural networks, including reservoir computing approaches. I would therefore have expected a better motivation for the particular choice, and an explanation what specifically is being measured early on in the paper.

The most widely used reservoir computing approach are Echo State Networks (Jaeger 2001), these are determinstic recurrent reservoirs. The work here appears to be concerned with stochastic reservoirs, in my view this choice is interesting but should be reflected in the title of the paper, but it not even mentioned in the abstract, and only in passing in the introduction.

I also find the title less informative than it could be, because it is not clear what limits are being investigated. For example there has been prior work on information processing and the memory capacity of echo state networks (eg Boedecker et al, 2012; Barancok et al, 2014). These approaches use information theoretic measurements. Other measures that are typically used for reservoir computing approaches are investigations of the spectral radius, or Lyapunov exponents (and their connection to other dynamical systems).

I find it difficult to see how the work helps in characterising the capability of a reservoir-based approach. In general, the capacity will increase as the number of neurons / basis-functions increases. The text and/or the chosen metric (IPC) in the work didn't help me to understand what the goal is, to either perfectly reproduce (memorize) a given signal, or to predict a given dynamical system based on some of its observations.

There appears to be only little coherent connections between the individual sections of the work and I find it hard to follow the logical flow in the work, or between the equations in the work. As a result, it is also hard to see what is the exact contribution of the work, and how it applies to research in reservoir computing methods.

The references are rather old and quite	incomplete.

**Questions:**

Why is IPC an appropriate measure for reservoir computing methods, and what specifically can we expect to learn from it?

---

### Meta-Review · Area_Chair_LLdn · 2023-12-06

**Metareview:**

**Summary**

This paper studies computational limitation of stochastic reservoir computers. More concretely, it provides a characterization of the information-processing capacity (IPC) of a physically-constrained stochastic reservoir that it is polynomial in the number n of bits (Theorem 2). An implication of the result in view of the hardness of reservoir learning is also discussed (Theorem 4).

**Strengths**

This work aims at providing insights about our understanding of abilities/limitations of reservoir computers.

**Weaknesses**
- All the reviewers felt a great deal of difficulty in understanding the contents and the contributions of this paper, implying that this paper lacks clarity in its presentation.
- The motivation of using the IPC as the measure for information-processing capability of reservoir computers is not well discussed.

**Justification For Why Not Higher Score:**

Because of the clarity issue raised by all the reviewers, it is difficult to imagine that this paper, if accepted, would benefit other researchers in the relevant field, at least in its current form. The authors did not submit their rebuttal, so that those criticisms raised by the reviewers have not been addressed properly.

**Justification For Why Not Lower Score:**

N/A

---

### Decision · Program_Chairs · 2024-01-16

Reject